# Immersive Virtual Reality as Analgesia during Dressing Changes of Hospitalized Children and Adolescents with Burns: A Systematic Review with Meta-Analysis

**DOI:** 10.3390/children7110194

**Published:** 2020-10-22

**Authors:** Yannick Lauwens, Fatemeh Rafaatpoor, Kobe Corbeel, Susan Broekmans, Jaan Toelen, Karel Allegaert

**Affiliations:** 1Department of Public Health and Primary Care, Academic Centre for Nursing and Midwifery, 3000 Leuven, Belgium; yannick.lauwens@hotmail.com (Y.L.); fatemeh.rafaatpoor@gmail.com (F.R.); kobe_corbeel@hotmail.com (K.C.); susan.broekmans@uzleuven.be (S.B.); 2Department of Development and Regeneration, KU Leuven, 3000 Leuven, Belgium; jaan.toelen@uzleuven.be; 3Department of Pediatrics, University Hospitals UZ Leuven, 3000 Leuven, Belgium; 4Department of Pharmaceutical and Pharmacological Sciences, KU Leuven, 3000 Leuven, Belgium; 5Department of Clinical Pharmacy, Erasmus MC, 3000 GA Rotterdam, The Netherlands

**Keywords:** children, adolescents, burns, dressing changes, virtual reality, pain

## Abstract

Children and adolescents with severe burns require medical and nursing interventions, associated with pain. As immersive virtual reality (VR) gained prominence as non-pharmacological adjuvant analgesia, we conducted a systematic review and meta-analysis on the efficacy of full immersive VR on pain experienced during dressing changes in hospitalized children and adolescents with severe burns. This exercise included quality and risk of bias assessment. The systematic review resulted in eight studies and 142 patients. Due to missing data, four studies were excluded from the meta-analysis. Fixed effects meta-analysis of the four included studies (n = 104) revealed a large effect size (ES) (Standardized Mean Difference = 0.94; 95% Confidence Interval = 0.62, 1.27; Z = 5.70; *p* < 0.00001) for adjuvant full immersive VR compared to standard care (SC). In conclusion, adjuvant full immersive VR significantly reduces pain experienced during dressing changes in children and adolescents with burns. We therefore recommend implementing full immersive VR as an adjuvant in this specific setting and population. However, this requires further research into the hygienic use of VR appliances in health institutions. Furthermore, due to the high cost of the hardware, a cost–benefit analysis is required. Finally, research should also verify the long term physical and psychological benefits of VR.

## 1. Introduction

Burn injuries are a major public health problem with a high mortality rate [1]. The incidence rate of burn injuries for children aged under sixteen is almost 25% [2]. Children under the age of five are commonly involved, likely due to their underdeveloped motoric and cognitive functions [2,3].

Severely burned patients require frequent painful interventions, such as dressing changes to prevent infection and improve wound healing [4]. The frequency of dressing changes depends on the amount of exudate and type of dressing material, and therefore varies between once a week, up to twice daily in hospitalized burn patients [5]. The process of dressing change is perceived by children to be the most distressing and painful moment during hospitalization [4,6,7].

Failure to sufficiently handle this pain results in physical and psychological effects [8]. Unfortunately, pain management during dressing change is often insufficient with the traditional methods, resulting in anxiety in children and their parents [9]. Other psychological effects include depression and (posttraumatic) stress disorder [8]. Regarding the negative somatic effects, studies documented that pain blunts wound healing [10,11]. Therefore, accurate pain management not only improves quality of life, but also reduces the aforementioned negative effects. Different pain management strategies are used, with (opioid) analgesics as the most commonly applied strategy [12]. However, reports show that the pain experienced by burn patients during wound care remains unacceptably high, even with the maximum allowed doses [13]. Additionally, the use of pharmacological analgesics in general causes side effects like nausea, vomiting, dizziness, respiratory depression, constipation, or tolerance. Physical dependence and addiction are also concerning [14].

Interestingly, distraction interventions as adjuvant therapies—together with opioids—hold the promise to decrease pain and anxiety in burn patients [8,15,16,17]. There are various types of distraction, with a range of passive (e.g., watching television or listening to music) or active (e.g., interactive toys, electronic games, or immersive technology) techniques [18]. Due to the rapid development of immersive technology in the last decade, there is a growing interest for this technology as distraction intervention. McDonnell (2019) [19] describes two technologies that are covered by the term “immersive technology”: augmented reality (AR) and virtual reality (VR). Poetker (2019) [20] describes AR: “AR technology adds layers of digital snippets on top of an existing environment, bridging the gap between the virtual and physical world.”. In other words, AR occurs in the real-world environment, where virtual objects are superimposed onto the real-world environment [21]. In contrast, VR is described as “a fully digital experience that can either simulate or differ completely from the real world.” [20]. Thus, the main difference between VR and AR is the environment where the activity takes place. Consequently, unlike VR, AR is not a fully immersive technology.

Previous studies compared immersive to non-immersive VR and reported that immersive VR is more effective in acute or chronic pain [22,23,24,25,26,27]. More specific, VR was more effective than video games in burn wound care [28]. Furthermore, it is also more effective than cognitive–behavioral techniques like watching a video or listening to music [29,30]. The participants’ stronger illusion of presence in VR games, through immersion and involvement by convergence of multisensory inputs, plays a crucial role in its effectiveness [13]. “The essence of immersive virtual reality analgesia is the patient’s illusion of going to a different place, the subjective experience of feeling present in the computer-generated world, as if the virtual reality world is a place they are visiting”, which indicates that VR works through an attentional mechanism of psychological pain distraction [31]. Mallari et al. (2019) [32] defined full immersive VR as “being engaged in a simulated world in the form of visual and auditory feedback with the ability to interact with and/or elicit a reaction from their virtual environment.” This further builds on the gate-control mechanism, which assumes that interpretation of pain intensity of an incoming signal depends on the patient’s focus, as the brain has a limited processing capacity [33,34]. Researchers argue therefore that when the patient’s attention is distracted by VR, less attention will be available for pain perception [23,28,31,35,36]. Therefore, we chose to only focus and investigate the effect of full immersive VR as most immersive technology.

Both hardware and software are required to use immersive VR. The hardware of traditional VR systems consists of a head-mounted device (HMD) with three-dimensional (3D) goggles, headphones, sensory input devices, and/or body tracking sensors [20,37,38]. The HMD creates an experience of movement through the virtual world by tracking the patients head movements [37]. This enables the user a multisensory experience. The software allows users to create different environments, commonly selected based on patient preferences. Furthermore, patients can adopt an interactive avatar to explore this virtual environment [38]. Different studies state that immersive VR has positive effects on pain in adults, children, and adolescents [9,28,39,40,41]. Several studies evaluated the effectiveness of VR during wound care in burn patients, and have demonstrated the effects of VR as adjuvant to pharmacotherapy [13,27,28,31,42].

This systematic review aims to provide an overview of the effect on pain of fully immersive VR in hospitalized children and adolescents with burn injuries during dressing changes. By conducting a subsequent meta-analysis, we aim to determine the effect size (ES) of immersive VR compared to standard care (SC). Such findings are fruitful to improve nursing care, pain management and to guide future research.

## 2. Materials and Methods

### 2.1. Eligibility Criteria

Articles had to document primary research and had to be published in peer reviewed journals in either English or Dutch, without limitation on publication date. Exclusion criteria: the application of non-immersive VR, semi-immersive VR, or other distraction interventions, any other kind of wound care besides burns and/or if results of children and adults were pooled and remained indistinguishable. Reviews, conference papers, and abstracts were also excluded. For inclusion in the meta-analysis, studies were eligible if the following data were available: a mean and/or median score —and a measure for dispersion—for pain during the procedure for both the immersive VR group (VRG) and the standard care group (SCG).

Participants were considered eligible if they were: burn patients aged 2–18 years, hospitalized, and undergoing wound care. Patients had to be able to score their own pain, with or without a visual tool. We chose to only include self-reported measures by children. For the subsequent meta-analysis, we only included validated pain scales that quantified the pain experienced. Burn patients were defined as patients with second and/or third-degree burns, without limitations based on the location of the burns.

As studies should evaluate the effect of full immersive VR on pain in children and/or adolescents with burns, at least one assessment with and one without the intervention was required. Restrictions on the intensity, duration, and/or frequency of the immersive VR intervention were not applied. Our focus was on outcomes assessed during dressing changes. When outcomes during dressing changes were not evaluated, we selected the moment of evaluation closest to the completion of the procedure.

### 2.2. Search Strategy

Following electronic databases were used to gather relevant scientific articles: PubMed, Embase and CINAHL. A list of search strategies and keywords for PubMed (Appendix A), Embase (Appendix A) and CINAHL (Appendix A) are provided. The PubMed search was conducted on 26 February 2020. Embase and CINAHL on 27 February 2020. Forward and backward snowball-methods were used to gather additional relevant articles. Both snowball methods were performed between 28 February and 5 March 2020, and were based on the results of the primary search. Finally, on 22 March 2020 a rerun of all searches was performed.

### 2.3. Data Collection

#### 2.3.1. Studies Selection

The primary search in PubMed, CINAHL and Embase resulted in 95 articles. One additional article was identified throughout the snowball-method. After excluding duplicates, 61 articles were retained. Article screening was conducted in two phases. During primary screening, these 61 articles were independently screened based on title and abstract. If there were any discrepancies, an independent researcher (A.K.) was asked to take the final decision. This resulted in the exclusion of 49 articles. During secondary screening, full-texts of the remaining twelve articles were read and assessed for eligibility. Articles were included if predetermined inclusion criteria were met. The full-text of each article was independently read and assessed by F.R., Y.L., and K.C. An independent researcher (A.K.) was asked to take the final decision in the event of discrepancies. This resulted in the inclusion of eight articles.

The rerun of the search strategy in PubMed, CINAHL and Embase yielded two new articles. All newly retrieved articles were evaluated and excluded. An extensive overview of the study selection process is shown in Appendix A.

#### 2.3.2. Data Extraction

For each article we tried extracting following study characteristics: author(s), year of publication, title, study design, number of participants, age, study population, moment of VR, duration of VR, VR equipment, VR environment, treatment conditions, comparison intervention, key findings, type and moment of outcome measure(s), and relevant statistical test results. Each author had to independently extract the data. These data were subsequently reviewed by a second author. The third author was available in the event of discrepancies. For the meta-analysis the various pain scale scores were converted to a scale of 100. We chose to convert the units of effect measure to calculate with a higher accuracy and to pool data for the meta-analysis. The main characteristics and results of studies retained in the systematic review were tabulated (Appendix B
Table A1, Table A2, Table A3 and Table A4).

### 2.4. Data Analysis

#### 2.4.1. Assessment of Study Quality

##### Quality of Evidence

The quality of each included article was independently evaluated by two researchers. This was done using either the Physiotherapy Evidence Database (PEDro) scale (Appendix B
Table A5) or the Modified Downs and Black index (MD & B) (Appendix B
Table A6). A PEDro score between 6 and 10 indicates high quality, a PEDro score of 4–5 is fair, while a PEDro score <3 reflects poor quality [43]. A MD & B score <14 reflects poor quality [44].

##### Assessment of Risk of Bias

The risk of bias was independently assessed by K.C. and Y.L. The third researcher (R.F.) was consulted if there were any discrepancies. Assessment was performed with either the Risk of Bias (RoB) 2 tool [45] or the Risk of Bias in Non-randomized Studies of Interventions (ROBINS-I) tool [46]. Using the online Risk of Bias Visualization (robvis) tool, we constructed different plots (Appendix B
Figure A1, Figure A2, Figure A3 and Figure A4) to list the assessed risk of bias per article [47].

#### 2.4.2. Synthesis of Results

##### Measures of Treatment Effect

The meta-analysis was performed using RevMan 5.3 [48]. The ES measure was calculated by comparing the difference in treatment. Where possible, Cohen’s d with a 95% confidence interval (CI) was calculated. We used the following guideline for interpreting the ES: 0.2 = small, 0.5 = medium and 0.8 = large effect [49]. Furthermore, heterogeneity was calculated using the I^2^ test, as the Chi^2^ test has a lower power for small sample size studies [50]. The following rule was used to assess heterogeneity: 25% = low, 50% = moderate, and 75% = high [51].

##### Dealing with Missing Data

If summary data were unavailable, we tried to calculate them with the available data or aimed to contact the corresponding author(s). If none of these methods worked, the article could not be included in the meta-analysis.

##### Data Synthesis

Data were included in this meta-analysis when the target population, intervention, and outcome of interest were in line with the inclusion criteria. For each study included, we calculated the standardized mean difference (SMD) and their 95% CI using a fixed effects model. Using RevMan, we visualized calculations with a forest plot. Chan et al. (2007) [13] did not mention the standard deviation (SD) of the reported pain intensity during dressing change. We could however calculate the SD using the reported standard error (SE).

## 3. Results

### 3.1. Description of Studies

We summarized the main characteristics and results of studies retained in the systematic review in Figure 1 and Appendix B
Table A1, Table A2, Table A3 and Table A4. Studies were conducted between 2000 and 2019. In all studies, adjuvant VR was compared to SC, except for Hoffman et al. (2000) [28] who used standard care in combination with a video game as comparison intervention.

#### 3.1.1. Excluded Studies

During secondary screening, the full-texts of the remaining twelve studies were read by at least two researchers. Among the twelve studies, four were excluded: two studies [53,54] did not have the target population and the other two articles [55,56] did not have the target intervention. Additionally, two articles [57,58] were found in the rerun, but were excluded.

For inclusion in the meta-analysis, studies required aforementioned summary data. Since some data were unavailable, we tried to contact the authors of the following articles: van Twillert et al. (2007) [59] and Jeffs et al. (2014) [60]. van Twillert et al. (2007) [59] replied that they were no longer able to share their original research data. Jeffs et al. (2014) [60] as well failed to grant access to the raw data. These two articles could therefore not be included in the meta-analysis. The studies of Hoffman et al. (2000) [28] (n = 2) and Scapin et al. (2017) [8] (n = 2) were excluded because of their study design.

#### 3.1.2. Included Studies

Eight studies remained included, reporting on 142 eligible participants. The study of van Twillert et al. (2007) [59] included 19 participants between 8 and 65 years, so that we only extracted data of participants <18 years (n = 6). The other seven studies included participants aged between 5 and 17 years [8,9,13,28,31,60,61]. Of these eight studies, four studies [9,13,31,61] were included in the meta-analysis. Details about the studies are summarized in Appendix B
Table A1, Table A2, Table A3 and Table A4.

Six studies used a randomized controlled trial (RCT) research design, of which four used a within-subject design and two a between-subject design. The two remaining studies were case series. In line with the selection criteria, all studies used immersive VR adjuvant to SC during wound care. However, different VR environments were used.

The visual analogue scale (VAS) was used in two articles [28,61] and the faces scale in three articles [8,9,13]. The visual analogue thermometer (VAT) was used in the article of van Twillert et al. (2007) [59]. VAT is an adaptation of the VAS which was developed to measure pain in patients with burns [62]. The article of Hoffman et al. (2019) [31] used the graphic rating scale (GRS) as a method of pain measurement. In the article of Jeffs et al. (2014) [60], a combination of adolescent pediatric pain tool (APPT) and word GRS (WGRS) was used. The WGRS is an adaptation of the GRS which was developed to measure pain in pediatric patients [63].

### 3.2. Study Quality Assessment

#### 3.2.1. Quality of Evidence

All included studies had generally moderate to high quality. For the six RCTs, five studies [9,13,31,59,60] were rated a PEDro-score of 7/11 and the study of Kipping et al. (2012) [61] a PEDro-score of 8/11. These scores can be explained because of allocation or non-availability of all data due to patient dropout. The two other articles [8,28] rated with MD & B, received both a rating of 15/27 (Appendix B
Table A5).

#### 3.2.2. Risk of Bias

##### Randomized Control Trials

The summary of risk of bias assessments of all included studies are shown in the Appendix B
Figure A1, Figure A2, Figure A3 and Figure A4. For bias related to randomization, one study was judged to have a "low" risk of bias due to adequate information [60,61]. Four studies [9,13,31,59] were judged to have "some concerns" of bias since a random allocation sequence was used, but no information was published on the allocation sequence concealment until enrolment and assignment. Blinding for this type of intervention (performance bias) is almost impossible due to the nature of the intervention. Although an independent assessor can be blinded. Thus, 2/8 studies [9,60] were single blinded. The study of Das et al. (2005) [9] analyzed data through a blinded assessor. In the study of Jeffs et al. (2014) [60] the key outcomes were collected through blinded study team members. The studies of van Twillert et al. (2007) [59] and Kipping et al. (2012) [61] indicated the difficulty of using double blinding but it was unclear if these and the remaining studies [8,13,28,31] used any blinding strategy.

A total of five studies [13,31,59,60,61] were judged to have a low risk of bias for incomplete outcome data due to the availability of the outcome of interest for all participants. The study of Das et al. (2005) [9] was rated as some concerns, because two participants were excluded related to side effects claimed to be caused by analgesics. Two studies [9,60] were judged low risk of bias for the outcome measurement as an appropriate method to measure the outcome of interest and unawareness of outcome assessors of the intervention were used. Three studies [13,59,61] were rated to have some concerns of bias because of their unclarity on the outcome assessors’ awareness of the intervention. Only one study [31] was judged to have a "high" risk of bias in outcome measurement as assessors were likely aware about the intervention. Finally, six studies were classified as [9,13,31,59,60,61] low risk of bias due to their preselected data analysis plan.

##### Non-Randomized Studies

ROBINS-I tool [46] was used to assess the quality in the two case series studies [8,28]. We rated these studies as low risk of bias for all eligibility criteria. Figure A3 shows the risk of bias across both case series studies.

The low risk of bias was due to no potential for confounding the intervention effect, participant selection was not based on characteristics observed after the start of intervention, a clear description of the intervention and there were no deviations reported from the intended intervention. The outcome of interest was provided for all participants and no systematic errors in intervention assessments were retrieved.

#### 3.2.3. Effect of the Intervention

##### Summary of Results for Studies Excluded from the Meta-Analysis

The two included case series [8,28], which each had two participants, reported that VR had a respectively relevant and considerable effect on pain reduction in children during wound care. In the study of Scapin et al. (2017) [8], the first patient scored his pain during SC 10/10 and during VR intervention 4/10. The second patient rated his pain level during SC 6/10 and during the VR intervention 0/10. In the study of Hoffman et al. (2000) [28], the first patient’s pain rating showed considerable reduction of 80 mm for the first VR session and 30 mm for the second VR session, compared to the control condition. The second patient showed a reduction of 47 mm in worst pain during VR intervention compared to the control condition.

In the study of van Twillert et al. (2007) [59] (n = 6), an overall tendency of remarkable pain reduction was reported in the VRG during dressing change. This comparison was made between the day before VR was offered, the day of VR and the day after VR. On the days before and after VR was offered, SC was provided. Moreover, the mean pain experienced was 2.90 (VRG) vs. 5.85 (SCG). The mean difference between the VRG and the SCG was −2.95 (*p* = 0.688). Furthermore, within-patient differences between the groups were significant (F = 13.2; *p* < 0.01).

The study of Jeffs et al. (2014) [60] (n = 28) showed that the VRG experienced less procedural pain than the SCG, but the difference was not statistically significant (difference = 9.7 mm; 95% CI = −9.5, 28.9; *p* = 0.32). Additionally, the estimated ES of self-reported pain scores between the VRG and the SCG was 0.535. In this study, it is also reported that the estimated procedural pain score (adjusted for various factors as age, sex, state anxiety, and pre-procedural pain, opioid drugs, and duration of treatment) is 28.7 (VRG) and 38.4 (SCG) [60].

The four studies included in this meta-analysis (n = 104) [9,13,31,61] found that VR adjuvant to SC reduced pain during dressing changes. Two out of four found that the effect was statistically significant (Das et al., 2007 [9], *p* < 0.01 and Hoffman et al., 2019 [31], *p* < 0.001). Pooling all four studies [9,13,31,61] was acceptable due to the homogeneity of the intervention and the study population. Moreover, the heterogeneity was calculated as moderate (I^2^ = 52%), confirming our decision to pool these studies [50]. Since the level of heterogeneity is moderate, we opted to use a fixed effects model. ES for self-reported pain could be generated for 4/8 studies [9,13,31,61]. Using the fixed-effects model, a large ES was found for VR on self-reported pain during dressing changes within our pre-specified range of age (SMD = 0.94; 95% CI = 0.62, 1.27; Z = 5.70; *p* < 0.00001) (Figure 2). This indicates a substantial clinical benefit. For further assessment of a publication bias, a funnel plot was used. Although our funnel plot does not include many studies, you can see there is some kind of symmetry present, which indicates or at least suggests absence of publication bias (Appendix B
Figure A5).

## 4. Discussion

The aim of this review with meta-analysis was to summarize the existing evidence, and investigate the efficacy of full immersive VR as analgesia for hospitalized children and adolescents during dressing changes for burns. We hereby identified eight studies (sample size ranged from 2 to 48; 142 patients included) to evaluate the effect of immersive VR on pain based on self-reported outcomes. The two included case series [8,28] also reported a substantial influence of VR on pain reported during dressing change in children and adolescents with burns. The meta-analysis consisted of 4/8 eligible studies [9,13,31,61]. According to Scheffler et al. (2018) [64], non-pharmacological interventions including VR have significant effects on pain relief during burn wound care in adults. Moreover, Birnie et al. (2018) [65], demonstrated the positive effect of VR distractions on reducing needle-related pain and distress in children. In line with these studies, this systematic review and meta-analysis states that VR has a reductive impact on the pain experienced among children and adolescents with burns during dressing changes compared to SC. However, there are still some remarks related to age range, timing of outcome assessment, analgesics administered, and blinding issues.

Previous research suggested that variables like age can impact treatment efficacy [64]. For example, Dahlquist et al. (2009) [65] implies that fully immersive VR is most effective in children between 10 and 14 years of age. In the included studies, the youngest hospitalized patient was five years old. This could be explained by the fact that children need to be able to express their pain verbally, which is difficult when they are younger. Therefore, children should be able to engage and interact with the VR stories to optimize the effect of VR [66].

The timing of outcome assessment may also influence VR efficacy. All included studies used VR during dressing changes. However, the exact moment of self-reported pain assessment varies between studies. Most of the studies [9,28,31,60] measured self-reported pain after wound care to avoid interrupting the intervention. Chan et al. (2007) [13], Kipping et al. (2012) [61] and Scapin et al. (2017) [8] evaluated self-reported pain before, during, and after dressing changes. In contrast, van Twillert et al. (2007) [59] only evaluated outcome measures during dressing changes. Furthermore, the timing of VR application within each patient’s hospitalization or healing process also varies among these studies. van Twillert et al. (2007) [59] looked at patients with an expected hospital stay for at least four days, where VR would be assessed within the first week of admission. Das et al. (2005) [9] compared VR against SC during the patient’s second or third dressing change. Chan et al. (2007) [13] looked at patients who were on their third to fifth day after their burn accident to compare VR with SC. Scapin et al. (2017) [8] and Hoffman et al. (2019) [31] did not mention these specific criteria. However, we could derive some information of the case series of Scapin et al. (2017) [8]. The first case got the intervention on day one, the second case on day eight after admission [8]. The study of Hoffman et al. (2000) [28] looked at patients’ first dressing change after skin graft surgery. Finally, Kipping et al. (2012) [61] and Jeffs et al. (2014) [60] used the intervention on patients during their first conscious dressing change. We could not yet retrieve data on the impact of repeated, intra-patient VR use.

The use of pharmacological analgesia in the included studies was poorly described. In the study of Chan et al. (2007) [13] paracetamol was used, while in the study of Scapin et al. (2017) [8] morphine was used. Unfortunately, this was the only information on analgesics provided. Therefore, the experience of pain among the included studies could be influenced by the use of different types or doses of analgesics.

Four out of six RCTs [9,13,31,59] adopted a within-subject design to eliminate differences in SCG and pain perception. However, this type of design and intervention, made it almost impossible to blind the participants for the VR intervention. Therefore, there will always be a risk of bias due to placebo effects. In addition to the patients, it is also nearly impossible to blind the assessors during the VR intervention, which increases the risk of performance bias.

This meta-analysis has some methodological limitations. Due to our specific research topic and strict criteria, we were only able to include eight studies in our review. Additionally, because most of these studies have relatively small sample sizes, it is difficult to generalize the outcome. Furthermore, we were unable to perform a subgroup analysis because of the small samples in the included studies. Although published guidelines for RCT-reporting are available [67], many authors still did not include all their summary statistics. The snowball-method was applied to 60 articles which were divided among the three reviewers. The absence of a second opinion during this process could have induced selection bias. Selection bias could also have arisen because we were not blinded for each other’s scores during the screening of articles.

The strengths of this review consist of a thorough and up to date literary search, which focused on a high quality of evidence. Six out of eight included studies were identified as RCTs and were evaluated with the PEDro-scale as high quality. Although case series are not considered as the highest quality of evidence, they were both evaluated as moderate to high quality using the MD & B-checklist. Furthermore, we used strict predetermined inclusion and exclusion criteria for eligibility in this review and meta-analysis.

With the result of our meta-analysis, we can state that VR does have a reductive impact on the pain experienced among children and adolescents with burns during dressing changes. Therefore, we suggest that VR should be used as adjuvant therapy in children with burns. However, a larger sample size can help to better understand the efficacy of VR as an adjuvant treatment for pain relief within this specific population. From a clinical research perspective, this also means that research should shift from single short term efficacy compared to placebo studies, to more repeated use or comparative approaches, cost-effectiveness, and safety studies. It is commonly suggested that VR requires expertise, time, and effort. However, it seems that VR systems can be applied without a lot of time and effort in clinical routine care [25]. This requires further research into the hygienic aspects of VR use in health institutions. As time efficiency is another key criterion of health care, research on this aspect is recommended. Furthermore, due to the hardware costs, these aspects should also be included in a cost-benefit analysis. Finally, research should also verify the long term physical and psychological effects of VR, including repeated use.

## 5. Conclusions

This review presents important information regarding the effectiveness of VR as an adjuvant therapy during wound care in children and adolescents. Adjuvant full immersive VR reduces pain during dressing changes in children and adolescents with burns, as shown in the results of our meta-analysis. Therefore, we recommend the use of VR in hospitalized children with burns. Furthermore, the present systematic review and meta-analysis could be used to guide future research and improve current practice among children and adolescents with burn injuries.

## Figures and Tables

**Figure 1 children-07-00194-f001:**
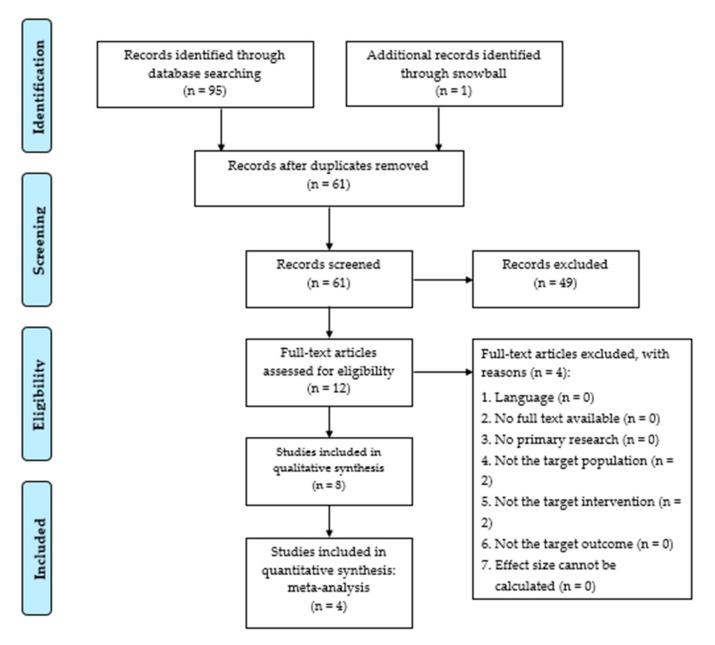
Flow diagram after rerun adapted from “Preferred Reporting Items for Systematic Reviews and Meta-Analyses: The PRISMA Statement.” by Moher D., Liberati A., Tetzlaff J. and Altman D.G., 2009, PLoS Med 6(7), p. 3 [52].

**Figure 2 children-07-00194-f002:**
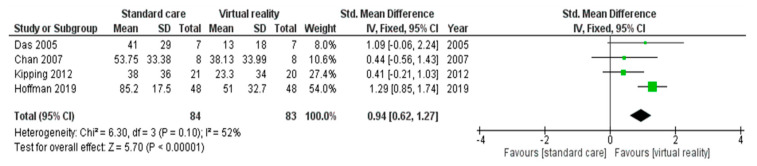
Forest plot: Fixed effects meta-analysis for the effect of virtual reality (VR) on self-reported pain during dressing changes compared to standard care (SC) [SD = standard deviation; IV = weighted mean difference; I^2^ = I-square heterogeneity statistic; CI = confidence interval; df = degrees of freedom; Z = Z statistic].

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
