# Peer review of "Immersive Virtual Reality as Analgesia during Dressing Changes of Hospitalized Children and Adolescents with Burns: A Systematic Review with Meta-Analysis"

_children, 2020, doi:10.3390/children7110194_

Round 1

Reviewer 1 Report

Immersive virtual reality as analgesia during dressing changes of hospitalized children and adolescents with burns: a systematic review with meta-analysis written by Lauwens et al. It is overall a good review but there are a few things/concerns need to correct before publication.  Why figures 1 and 2 repeated first in the text and then with different data at the end of the paper.  Figures 2-5 need improvement in terms of text font of X- and Y-axis to clear readability.  Write the date properly for reference throughout the review. 

Author Response

Why figures 1 and 2 repeated first in the text and then with different data at the end of the paper

Figure 1 provides the commonly reported ‘flow diagram’ as instructed in the PRISMA statement, while table A1 provides a detailed overview on the included studies, and the Supplement tables (tables S1- S4) provides the full overview on search terms. In our opinion, this information is indeed based on the same source, but more detailed for the reader with specific interest on the methodological approach as conducted. The same holds true for Figure 2, as we strongly feel that the forest plots does provide a condense overview of the final results, while the detailed information is indeed provided in the appendix and the supplement.

Figures 2-5 need improvement in terms of text font of X- and Y-axis to clear readability

we have adapted these figures

Write the date properly for reference throughout the review. 

We have harmonised the date throughout the document (full version, appendix, supplement and references)

Reviewer 2 Report

I feel that the paper is well-done. I would suggest not using the term "victims" on line 40, since it is not preferred by the burn survivor community. 

Author Response

I feel that the paper is well-done. I would suggest not using the term "victims" on line 40, since it is not preferred by the burn survivor community. 

we have adapted this sentence in the introuduction to: 

Children under the age of five are commonly involved